# Privacy *Déjà Vu* Effect: Resurfacing Sensitive Samples in Continual Fine-tuning

## Abstract

Continual fine-tuning of large pre-trained models is now ubiquitous in industry for adapting a model to freshly collected user data. Existing privacy protection practices assume earlier training data is less sensitive and thus focus on the latest arriving samples. We challenge this assumption by tracking per-sample membership-inference risk across sequential fine-tuning rounds of popular transformer-based models, ViT for image data, and BERT for text data. Our experiments reveal the *Privacy Déjà Vu Effect*: new data can *remind* the model of semantically similar legacy samples, possibly elevating their privacy risk significantly. We further demonstrate that this resurgence is closely correlated with the latent-feature-space similarity between old and new examples. These findings underscore the need for a more comprehensive privacy protection mechanism in continual fine-tuning. We have published our code at `https://anonymous.4open.science/r/Privacy-Deja-vu-Effect-F006/README.md`.

## 1 Introduction

As machine learning models continue to grow in size, training them from scratch becomes prohibitively expensive. Many companies, such as Reddit (Reddit, Inc.) and IBM (Stapleton), instead opt to fine-tune pre-trained base models using newly arriving data samples incrementally. Such continual fine-tuning (Wang et al., 2024) is a common strategy employed in sectors such as customer support, recommendation systems, and autonomous vehicles, where new user data is constantly integrated to enhance the model performance and freshness.

However, privacy concerns arise in this practice. A widely held assumption is that the most recent training data predominantly influence the model's behavior, and the old data likely suffers from catastrophic forgetting, resulting in a largely reduced ability to capture the utility of old samples (Wang et al., 2024). Thus, the common belief is to prioritize privacy safeguards for new samples while progressively neglecting older ones to reduce the protection expenses (Chathoth et al., 2022a; Desai et al., 2021; Chathoth et al., 2022b). Correspondingly, industry parties start to adopt this strategy. IBM Watsonx, for instance, implements robust privacy measures on the latest fine-tuning samples (IBM Cloud, 2023). Regulatory documents, e.g., EDPB Opinion 28/2024 (European Data Protection Board, 2024) also stress that newer data in continual fine-tuning should be protected primarily.

Yet, our research reveals a counterintuitive phenomenon which we call "Privacy Déjà Vu Effect": While the standard continual fine-tuning is applied to make models more likely to forget the old distribution (Bafghi et al., 2024), some old samples, initially exhibiting low privacy sensitivity, become more vulnerable during model fine-tuning. It seems new samples remind the model of its old memory.

**Scope of Our Research.** In this paper, we show that such Privacy Déjà Vu Effect exists in the fine-tuning processes on two typical transformer-based models, ViT and BERT, covering image- and text-based applications, respectively. To detect the change of samples' privacy risks, we consider a canonical family of privacy attacks called membership inference attacks (Shokri et al., 2017), which predict whether or not a given example is contained in the model's training set. Privacy risk measures derived from membership inference attacks, such as the per-sample $\frac{\text{TPR}}{\text{FPR}}$ ratio, an empirical measure conceptually similar to the privacy budget $e^\epsilon$ in $(\epsilon, \delta)$-differential privacy, have been widely accepted as indicators of practical privacy risks (Aerni et al., 2024; Jagielski et al., 2020; Nasr et al., 2023; Steinke et al., 2024). By using this privacy risk measurement in consecutive fine-tuning ViT models

on Tiny-ImageNet-200 and BERT models on IMDb, we observed that the largest increase in privacy risk for sensitive samples exhibits an astonishing jump.

We perform several experiments to understand the root causes of the Privacy Déjà Vu Effect. This effect can be partially explained by the level of distribution shifting between neighboring fine-tuning rounds - if the new samples focus on a few classes (not randomly sparsely distributed), the privacy risk of the old samples in similar classes gets boosted more than other samples. An in-depth analysis of the similarity between consecutive steps of fine-tuning confirms that old samples similar to the newcomers are more prone to a privacy risk surge. Thus, the name Privacy Déjà Vu precisely captures how the new samples revive the model's memory about similar old ones.

This Privacy Déjà Vu Effect challenges the currently adopted practice for economical privacy protection: *protecting mainly the newest samples in fine-tuning* is not sufficient. Our work shows that such biased protection may introduce significant privacy leakage to the related old samples.

**Contributions.** (1), We reveal the Privacy Déjà Vu Effect: new data in continual fine-tuning can increase the privacy risk of previously safe samples. (2), Experiments on two representative foundation models and two benchmark datasets show that the effect might commonly exist. (3), We have also experimentally studied the reasons behind this effect and identified the significant factors.

## 2 RELATED WORK

**Continual Fine-tuning and Catastrophic Forgetting.** Unlike conventional machine-learning pipelines that assume a static data distribution, continual learning adapts to non-stationary streams of data (Wang et al., 2024). Its central challenge is *catastrophic forgetting*: updating on new data degrades performance on previously learned tasks (McClelland et al., 1995; McCloskey & Cohen, 1989). Gido et al. show that the same problem arises during continual fine-tuning of neural networks (Van de Ven & Tolias, 2019). To curb forgetting, Hadsell et al. propose preserving weights that are critical for early-stage data (Hadsell et al., 2020). Industry systems adopt additional heuristics such as hard attention to historical samples (Serra et al., 2018). The REMIND approach rehearses compressed representations of past data (Hayes et al., 2020). Focusing on transformer models, Bafghi et al. report that full-parameter continual fine-tuning suffers the most severe forgetting (Bafghi et al., 2024).

**Privacy Risks in Continual Fine-tuning.** Because catastrophic forgetting appears to reduce the model's memory of earlier data, some studies argue that it can reduce the privacy risk of old data, and therefore we should concentrate protection on newly ingested samples (Wang et al., 2025; Chathoth et al., 2022a; Desai et al., 2021; Chathoth et al., 2022b). For instance, Hassanpour et al. assign smaller differential-privacy budgets to successive training rounds (Hassanpour et al., 2022). However, other work demonstrates that legacy data can remain susceptible to extraction attacks even after multiple fine-tuning rounds on both vision and language models (Jagielski et al., 2023; Chen et al., 2024; Borkar et al., 2025). These findings challenge the assumption that older data can be neglected in privacy analyses for continual fine-tuning systems.

## 3 PRELIMINARIES

In this section, we introduce a typical continual fine-tuning method on foundation models. We also describe the process of measuring privacy risks using a membership inference attack, specifically the Offline Likelihood Ratio Attack (LiRA).

**Continual Fine-tuning on Foundation Models.** Let $\mathcal{D}_k$ denote a data distribution and $S_k = \{(x_{i,k}, y_{i,k})|(x_{i,k}, y_{i,k}) \in \mathcal{D}_k, i = 1 \dots N_k\}$ denote a training dataset of $k$-th round fine-tuning, and $f_k \leftarrow \mathcal{T}(S_k)$ denote the model we obtain by fine-tuning the previous model $f_{k-1}$. Continual fine-tuning can be categorized into various types according to the difference among $\mathcal{D}_k$ (Wang et al., 2024). In this paper, we consider domain-incremental fine-tuning, where each $\mathcal{D}_k$ has the same label space but possibly different distributions, as considered by previous works (Jagielski et al., 2023; Carlini et al., 2022b). This setting also facilitates the in-depth study of causes of the Privacy Déjà Vu Effect, which will be introduced in Section 4.

As shown in Figure 7 in Appendix 1, the $k$-th round model $f_k$ is given by fine-tuning $f_0$ sequentially with $\{S_1, ..., S_k\}$, where $S_0$ is the dataset to train a foundation model $f_0$, e.g., a ViT model pre-trained on the ImageNet (Dosovitskiy et al., 2021). Furthermore, we adopted the setting that fine-tunes all parameters (Bafghi et al., 2024). This strategy is most likely to catastrophically forget old distribution and thus considered to benefit the privacy protection of old data (Wang et al., 2025).

**Per-sample Privacy Risk Metric.** There are several metrics to estimate the per-sample privacy risk, e.g., per-sample attack success rate of Membership Inference Attacks (MIAs) (Carlini et al., 2022c) and the Fisher information of samples (Farokhi & Sandberg, 2017). We adopt the per-sample $\frac{\text{TPR}}{\text{FPR}}$ of an MIA as the privacy risk metric because it intrinsically relates to differential privacy (Aerni et al., 2024; Tramer et al., 2022) as follows.

Recall the definition of $(\epsilon, \delta)$–differential privacy (DP) (Dwork, 2006) for a randomized mechanism $\mathcal{M}$ acting on adjacent datasets $D, D'$. Assuming $\forall \mathcal{O} \subseteq \text{Range}(\mathcal{M})$:
$$\Pr[\mathcal{M}(D) \in \mathcal{O}] \le e^\epsilon \Pr[\mathcal{M}(D') \in \mathcal{O}] + \delta.$$

When $\delta \approx 0$ ($\delta$ is always very small in practice), the DP guarantee in the *hypothesis-testing* form ensures no sample's MIA result $\frac{\text{TPR}}{\text{FPR}}$ exceeds the privacy budget (Kairouz et al., 2015; Dong et al., 2022):

$$\frac{\text{TPR}}{\text{FPR}} \le e^\epsilon$$

where TPR and FPR denote, respectively, the true and false positive rates of any distinguishing attack that decides whether the output $\mathcal{M}(\cdot)$ came from $D$ or $D'$. Due to the statistical nature of machine learning, even without a DP randomization mechanism, an MIA on a non-DP model still gives a measure $\frac{\text{TPR}}{\text{FPR}}$ for each sample, which we consider the sample's "inherent privacy risk". While an ideal attack can precisely estimate this risk measure, in practice, we can only use the best MIA so far to get the maximum $\frac{\text{TPR}}{\text{FPR}}$ estimate. We choose to use one of the most powerful MIAs, LiRA (Carlini et al., 2022a), in the experiments. Samples with larger $\frac{\text{TPR}}{\text{FPR}}$ values are considered more risky.

**The Likelihood Ratio Attack (LiRA).** LiRA is considered one of the most powerful MIAs. Thus, we use LiRA as the backbone attack of our privacy risk estimator. There are online and offline versions of LiRA, which are based on two-sided and one-sided hypothesis testing, respectively. Offline LiRA is more efficient because it only needs to estimate the "out" distribution by sacrificing some marginal effectiveness. For simplicity, we use the term "LiRA" to represent the offline version of LiRA in this paper. The details of LiRA are as follows.

1. Estimate distribution of "out" logits. Given a machine learning model $g$ and the training strategy $\mathcal{G}(\cdot)$, LiRA first train multiple "shadow models" $\{g_j \leftarrow \mathcal{G}(X_j)\}$ on random subsets $\{X_j | X_j \subset \mathcal{D}, j = 1..m\}$ drawn from the known training data distribution $\mathcal{D}$. For any non-member sample $(x, y)$, e.g., for $g_j$, LiRA computes the logits of the confidence of the target class $y$, $p = g_j(x)_y$: $\log \frac{p}{1-p}$. This computation will be applied to a sufficient number of randomly selected non-members for $\{g_j\}$, respectively. The distribution of the logits values is approximately a univariate Gaussian distribution, the parameters of which can be estimated with these samples. We train 256 shadow models to estimate the distribution parameters, as suggested by (Carlini et al., 2022a).

2. Attack target sample $(x_i, y_i)$. To predict whether a target sample $(x_i, y_i)$ is a member of $X_j$, LiRA computes the logit transformation of the prediction of $g_j$, and computes the likelihood of the sample drawn from the "out" Gaussian distribution, denoted as $q$. In practice, there is a threshold $\tau$ for the adversary to classify the sample as an "in" or "out" sample. If $q < \tau$, then the prediction is "member", and "non-member" otherwise. Choice of $\tau$ will be introduced in the next subsection.

**Estimation of Per-sample $\frac{\text{TPR}}{\text{FPR}}$.** To estimate privacy risk $\frac{\text{TPR}_i}{\text{FPR}_i}$ of each sample $(x_i, y_i) \in X$, we conduct a random sampling of the training data $X$ to generate $m$ sample sets, where each sample set $X_j$ is generated by selecting each sample $x_i$ in $X$ with probability of 0.5. Each of the sample set is used to train a model $g_j$. Thus, $x_i$ is used by about $m/2$ models in training, which forms the ground-truth of the $x_i$'s membership in the $m$ models. We then use LiRA to compute $q_{i,j}$, the probability of sample $x_i$ is drawn from "out" Gaussian distribution. In our estimation, each threshold $\tau$ will give a pair of TPR and FPR by comparing the attacking results and the ground truths, and we choose the $\tau_i$ that gives the greatest $\frac{\text{TPR}_i}{\text{FPR}_i}$ for sample $x_i$.

## 4 METHODS

Privacy Déjà Vu Effect means a new fine-tuning round will expose the privacy of some training samples in the previous fine-tuning rounds. For simplicity, we will look at the change in privacy risk of a sample in $S_k$ after the model is fine-tuned on $S_{k+1}$. Next, we will discuss the methods we use to explore this effect.

### 4.1 MODEL AND DATASETS

We start with our choice of datasets and models.

**Datasets.** We adopt two standard benchmarks: Tiny-ImageNet-200 (Le & Yang, 2015) and the IMDb Large Movie Review corpus (Maas et al., 2011). To meet the domain-incremental setting in Section 3, on each stage, fine-tuning sets shares the same label set but differs in input distribution. We therefore merge original fine-grained labels into superclasses. Formally, let $\mathcal{C}$ be the original class set and $\mathcal{C}' = \{s_1, \ldots, s_J\}$ a partition of $\mathcal{C}$. Tiny-ImageNet provides $|\mathcal{C}| = 200$ classes grouped into $|\mathcal{C}'| = 22$ semantic clusters (e.g., "Vehicle" superclass spans original classes "limo", "sportscar", "wagon", etc. (Deng et al., 2009)). IMDb's ten rating buckets collapse into Neg=$\{1\text{–}4\}$ and Pos=$\{7\text{–}10\}$ (Maas et al., 2011). For Tiny-ImageNet-200, we sample two fine-grained classes per superclass to form $S_k \subset \mathcal{D}$; for IMDb, we sample one rating per superclass.

**Models.** Experiments cover two foundation architectures: ViT-B/16 pretrained on ImageNet-21k (Dosovitskiy et al., 2021) and BERT-base (uncased) (Devlin et al., 2019). We use the strategy introduced in (Jagielski et al., 2023) to mimic domain-incremental fine-tuning: each model undergoes two fine-tuning rounds: round 1 yields $f_k$, round 2 yields $f_{k+1}$, i.e., $f_1$ and $f_2$. Both rounds use the superclasses as labels, i.e., ViTs fit 22-classification tasks and BERTs fit binary-classification tasks. The validation sets are drawn from the entire validation set according to the superclasses and classes in the training sets. For instance, if superclass "Vehicle" in $S_k$ contains "Limo", then the corresponding validation set also contains "Limo". We adopt very small learning rates ($3 \times 10^{-6}$ for ViT, $1 \times 10^{-7}$ for BERT) and early stopping after three epochs without validation-loss improvement, storing the best checkpoint—as in (Jagielski et al., 2023). This protocol attains 93.4% validation accuracy on Tiny-ImageNet-200 and 94.2% on IMDb. In this paper, we fine-tune 500 $f_k$ ($m = 500$) to generate statistically significant estimation, which is suggested by (Gu et al., 2024). It takes 16.3 hours on 15 RTX-2080 Ti GPUs to finish all fine-tuning stages on ViT models, and 6.8 hours on BERT models. All experiments in the paper takes around 900 hours.

### 4.2 ESTIMATING PRIVACY RISK CHANGE

To estimate the per-sample privacy risk change of samples in $S_k$, we need to compute the per-sample privacy risk of samples in $S_k$ on model $f_k$ and $f_{k+1}$. As shown in Figure 7 in Appendix 1, this estimation has three steps:

1. Estimation over $f_k$: Following the LiRA-based per-sample $\frac{\text{TPR}}{\text{FPR}}$ estimation described in Section 3, and also as shown in Figure 7 in Appendix 1, we fine-tune $f_{k-1}$ to get $m$ models $\{f_{j,k} \leftarrow \mathcal{T}(S_{j,k}) | j = 1, \ldots, m\}$. Each sample in $S_{j,k}$ is randomly sampled with a probability of 0.5 from $S_k$. Thus, around $\frac{m}{2}$ datasets contain sample $x_{i,k} \in S_k$. And for each sample-model pair, we have the ground truth membership. We then use LiRA to attack the $x_i$ on each of the models to get $m$ predicted memberships, with which we can compute the sample-level privacy risk $R_{i,k} = \frac{\text{TPR}_{i,k}}{\text{FPR}_{i,k}}$ of $x_i$ on $f_k$.

2. Estimation over $f_{k+1}$: To estimate the privacy risk of each sample in $S_k$ after the $(k+1)$-th round of fine-tuning, we simply fine-tune each $f_{j,k}$ on the fine-tuning set $S_{k+1}$ and use the $m$ fine-tuned models $f_{j,k+1}$ to estimate the privacy risk $R_{i,k+1} = \frac{\text{TPR}_{i,k+1}}{\text{FPR}_{i,k+1}}$ of each sample $x_i \in S_k$ as step 1.

3. Estimating privacy risk change: For sample $x_i \in S_k$, we compute the privacy risk change as $\Delta_i = R_{i,k+1} - R_{i,k}$. A greater positive $\Delta_i$ indicates the $x_i$'s privacy risk is enhanced more in $f_{k+1}$. A negative $\Delta_i$ means the sample becomes safer.

## 4.3 STUDY THE PRIVACY DÉJÀ VU EFFECT

**Fine-tuning Strategies.** To assess the widespread nature of the Privacy Déjà Vu Effect, we implement two contrasting data update strategies: *SGD-New* and *SGD-Full*, as introduced (Jagielski et al., 2023). In *SGD-New*, the dataset $S_{k+1}$ comprises only new samples, randomly drawn from $\mathcal{D} \setminus S_k$, with a size set to half of $|S_k|$ to avoid the impact of dataset size difference between $S_{j,k}$ and $S_{k+1}$—specifically, $|S_k| = 30,000$ for Tiny-ImageNet-200 and $|S_k| = 15,000$ for IMDb. In *SGD-Full*, $S_{k+1}$ includes both the new samples and all data from $S_k$, effectively duplicating the previous dataset.

Figure 1 illustrates that, for both BERT and ViT models, and under both strategies, certain samples in $S_k$ exhibit increased privacy risk after fine-tuning. Notably, the *SGD-New* strategy results in a more pronounced Privacy Déjà Vu Effect compared to *SGD-Full*. We hypothesize that this difference arises because duplicating old data in *SGD-Full* reduces the sensitivity of some samples, thereby mitigating privacy risks. This observation aligns with findings by (Carlini et al., 2022c). We further validate this hypothesis in Section 5.2.

These findings confirm that the Privacy Déjà Vu Effect is prevalent in practice and appears to be influenced by the relationship between old and new data. Because the *SGD-New* setting avoid the impact of overlap between $S_k$ and $S_{k+1}$, to analyze the details of the effect, subsequent sections will focus on the *SGD-New* setting.

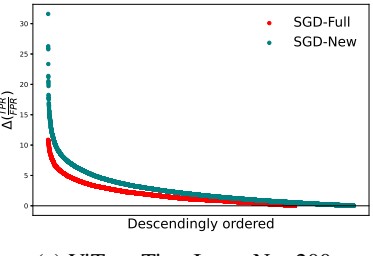

(a) ViT on Tiny-ImageNet-200

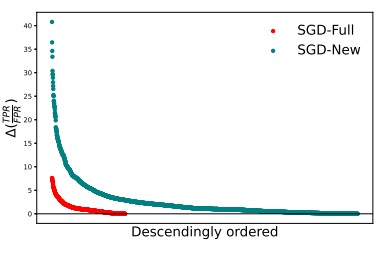

(b) BERT on IMDb

Figure 1: Descendingly ordered $\Delta$ with different continual fine-tuning strategies. Note that each set of results is sorted independently. Both models show that SGD-New causes more intense privacy risk increases for old samples. We omit samples with $\Delta(\frac{\text{TPR}}{\text{FPR}}) <= 0$, which cover roughly 68% of the ViT cases and 62.5% of the BERT cases.

**Simulating Distribution Shifting.** The results in Figure 1 suggest that Privacy Déjà Vu Effect intensifies with the correlation between the old data $S_k$ and the new data $S_{k+1}$, which is related to subpopulation distribution shifting. As shown by (Jagielski et al., 2023), simulating the subpopulation distribution shifting in continual fine-tuning is a proper way to study the correlation among data. We use the BREEDS framework (Santurkar et al., 2021) to simulate the subpopulation shifting, which works with a hierarchy of classes and samples similar classes in the superclass.

For ViTs, we first build $S_k$ by sampling two classes per superclass from Tiny-ImageNet-200 and train the $f_k$. For BERTs, we randomly choose one class per superclass from IMDb. To simulate a focused subpopulation shifting within a target superclass $s^\star \in \mathcal{C}'$, define the remaining pool of classes $C_{\text{rem}} = \{\mathcal{C} \setminus \{\text{classes in } S_k\}\}$ and let $n = |\{C_{\text{rem}} \cap s^\star\}|$. Then we form $S_{k+1}$ by sampling examples from $\alpha \times n$ classes chosen at random from $\{C_{\text{rem}} \cap s^\star\}$, where $\alpha \in (0, 1]$ controls shifting strength. We choose $\alpha$ from $\{0.2, 0.4, 0.6, 0.8\}$ for Tiny-ImageNet-200 and $\{0.4, 0.7, 1\}$ for IMDb. Fine-tuning $f_k$ on $S_{k+1}$ thus implements a BREEDS-style subpopulation shifting in the target superclass $s^\star$, isolating its impact on the Privacy Déjà Vu effect. In our experiments, we repeat experiments by trying each superclass as the target superclass and conclude our results.

## 4.4 ANALYZING THE DEJAVU EFFECT WITH DIFFERENT DATA SIMILARITY MEASURES

As the Privacy Déjà Vu Effect is similar to how people recall similar memories, we use two sample-to-sample similarity measures between the samples $S_k$ and $S_{k+1}$ to study the factors that cause the effect: **Structural Similarity (SSIM) (Wang et al., 2004)** – a perceptual, pixel-space measure that reflects how humans compare images. **Gradient Dot-Product (NTK-similarity) (Jacot et al.,**

**2018)** – a parameter-space measure that reflects how the model perceives the similarity between two samples.

**SSIM.** SSIM applies only to images, so we use it on ViTs. For every image $x \in S_k$, we compute its mean SSIM against $S_{k+1}$. We sort the images by their SSIM score and split them into 10 equal-sized quantiles. The quantiles with greater ID indicate that the old data is more similar to the $S_{k+1}$.

**NTK-similarity.** The neural tangent kernel compares inputs by the angle of their parameter-gradient "fingerprints." High similarity implies that training on one example produces a large first-order effect on another. We show the detailed computation of the NTK-simlarity in Appendix. Analogous to SSIM, we sort samples in $S_k$ by the NTK-similarity and divide into 10 equal-sized quantiles.

## 5 RESULTS

In this section, we study the two questions through our experimental results: (1) How does the Privacy Déjà Vu Effect perform in subpopulation distribution shifting scenarios? (Section 5.1) (2) What are the causes of the Privacy Déjà Vu Effect? (Section 5.2)

### 5.1 PRIVACY DÉJÀ VU EFFECT IN SUBPOPULATION DISTRIBUTION SHIFTING

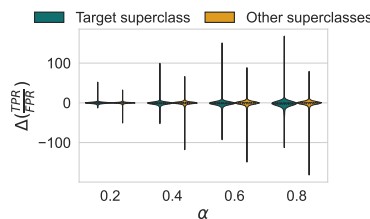

(a) ViT on Tiny-ImageNet-200

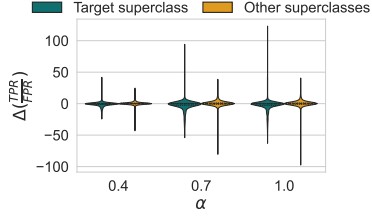

(b) BERT on IMDb

Figure 2: Violin graph of privacy risk change $\Delta$ of multiple strength of distribution shifting. The vertical line indicates the range of privacy risk change $\Delta$ across samples, and the bulge indicates the density. Results of the target superclass show samples with larger privacy risk increases than samples in other superclasses. Greater $\alpha$ causes more significant Privacy Déjà Vu Effect.

**Trend with Distribution Shifting Srength.** We show how the subpopulation shifting strength $\alpha$ impacts the Privacy Déjà Vu Effect in Figure 2a. We use the violin graph to more intuitively understand the impacts of fine-tuning on privacy risk changes. The vertical span shows the full range of the $\Delta$. Each violin shape represents the probability density over the y-axis. The narrow shape means most points are around zero (no dramatic privacy risk change). As the parameter $\alpha$ increases, the overall range of privacy change $\Delta$—as indicated by the vertical span of the violins in Figure 2—is larger for both the target superclass and other superclasses, in both BERT and ViT models. The pronounced bulges near $\Delta \approx 0$ reflect that many samples experience little change in privacy risk.

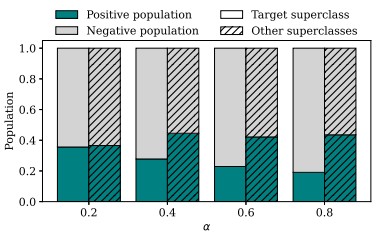

(a) ViT on Tiny-ImageNet-200

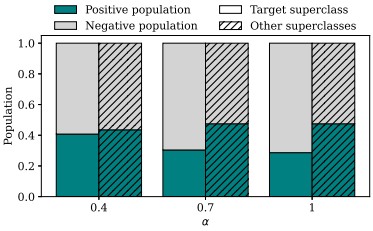

(b) BERT on IMDb

Figure 3: The privacy risk changes for the samples in the target superclass are more affected by the distribution shifting strength $\alpha$. With stronger shifting (larger $\alpha$), less samples in target superclasses show positive privacy risk change $\delta$, while which in other superclasses do not show obvious trends. We also show the results with different $\alpha$ in Appendix 3.

**Fraction of Positively Changed Samples.** To better understand this pattern, we inspect the population of samples with positive or negative $\Delta$. We quantify this by:

$$r^+ = \frac{|\{\, x \in S_k : \Delta(x) > 0 \,\}|}{|S_k|}, \qquad r^- = 1 - r^+.$$

where $r^+$ indicates the population of the samples that become more risky and $r^-$ indicates the population of safer samples in $S_k$.

Figure 3 shows that for other superclasses, $r^+$ converges to about $0.5$ as $\alpha$ increases, whereas for the target class, $r^+$ decreases. In other words, the Privacy Déjà Vu Effect in the target superclasses only shows in a smaller number of highly vulnerable examples, not a uniform change across all samples.

Combining Figure 2 and 3, the phenomenon is a striking analogy with human memory: just as a few new examples can cue recall of related past items, a small fine-tuning set "reminds" the models of old samples. Imagine a person who glimpses a handful of cars drawn from many makes (small $\alpha$); they instinctively recall vehicles across all brands. But if they then see dozens of cars from a single marque (large $\alpha$), their recall narrows to that one brand and neglects other brands (the vertical span). This metaphor inspired us to understand the Privacy Déjà Vu Effect from the perspective of similarity between old and new data. Intuitively, the new data will remind the model of some similar old data.

## 5.2 CAUSES OF THE PRIVACY DÉJÀ VU EFFECT

In this section, we study how the Privacy Déjà Vu Effect relates to data similarity.

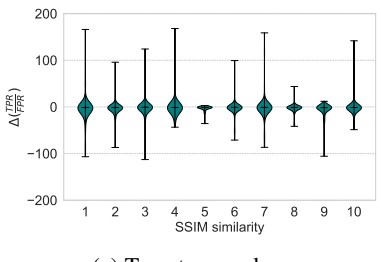
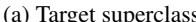
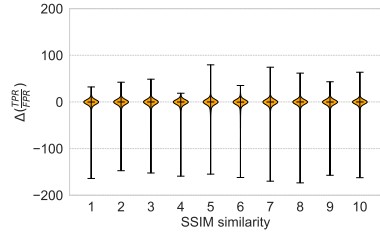

(a) Target superclass           (b) Other superclasses

Figure 4: Violin graph of $\Delta$ within ascendingly ordered SSIM quantiles when $\alpha = 0.8$. The rightmost is the quantile with the greatest similarity. The results do not show obvious relationship between SSIM similarity and the Privacy Déjà Vu Effect.

**Structural Similarity (SSIM) vs. $\Delta$.** Following the method in Section 4.4, we begin by asking whether raw image–level similarity between old and new samples correlates with the privacy risk change $\Delta$ in ViT models. In Figure 4a and Figure 4b, we draw the violin graph of $\Delta$ within each SSIM quantile when $\alpha = 0.8$. If higher visual similarity drove larger privacy risk changes, one would expect a higher $\Delta$ in the quantiles with greater ID. However, both panels exhibit no clear upward trend—raw image-based SSIM fails to predict which samples exhibit large $\Delta$. When we try various $\alpha$, there is also no obvious correlation between SSIM and $\Delta$. We show the results in Appendix 3.

Does this observation indicate that similarity is not correlated to $\Delta$ variability? Our answer is no. The key problem is the mismatch between *human-perceptual* and *model-perceptual* similarity: SSIM mimics our visual judgments, whereas ViTs base their decisions on latent feature representations.

**NTK-similarity vs. $\Delta$.** As introduced in Section 4.4, NTK-similarity captures the similarity from the perspective of machine learning model. Figure 5 shows the violin graph of $\Delta$ versus ascending NTK-similarity quantiles for $\alpha = 0.8$ (ViT) and $\alpha = 1$ (BERT). Figure 5a exhibits a clear upward trend: in the ViT models, the higher NTK similarity leads to more dramatic change of risks, including both larger increases and larger decreases for some sample in the target superclass. In the other superclasses, Figure 5b and 5d show downward trends, which means that higher NTK-similarity in other superclasses mitigates the sensitivity of old data. This is because they have a different label, which is a typical example of catastrophic forgetting in continual fine-tuning (De Lange et al., 2021). That is, the model $f_{k+1}$ will re-link new data to a label, which will impact the model's performance on some similar data in $S_k$ with different labels. For instance, assume a blue sport car image in $S_k$ was labeled as "sport car" and then the label was changed to "blue" in $S_{k+1}$, $f_{k+1}$ will link the data

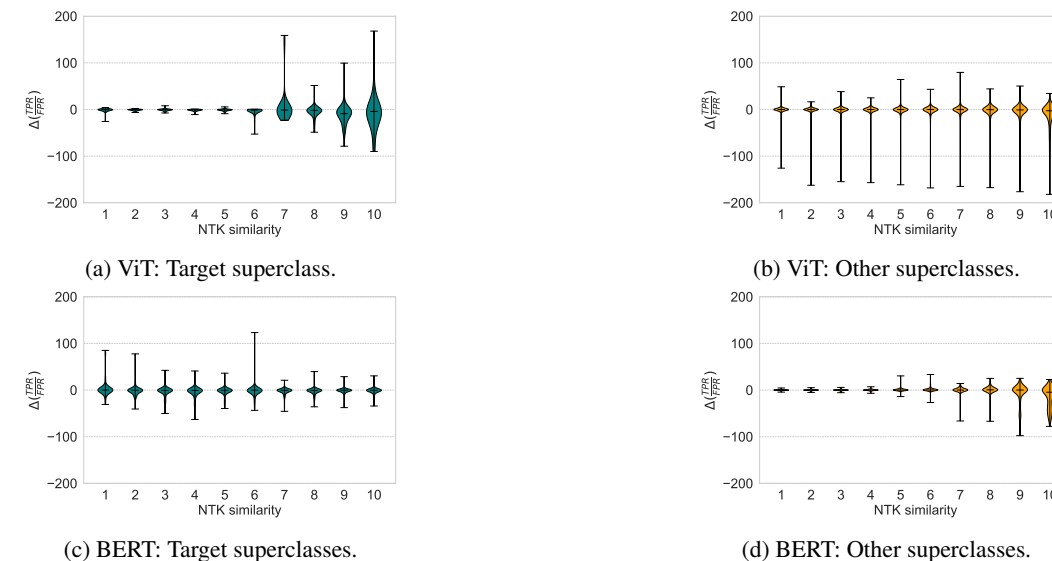

(a) ViT: Target superclass.  (b) ViT: Other superclasses.

(c) BERT: Target superclasses.  (d) BERT: Other superclasses.

Figure 5: Violin graph of $\Delta$ within ascendingly ordered NTK-similarity quantiles. The rightmost is the quantile with the greatest similarity. ViT: $\alpha = 0.8$; BERT: $\alpha = 1$. The Privacy Déjà Vu Effect show clear correlation with NTK-similarity in ViTs. However, the trends is abnormal in BERTs. We visualize the samples in Tiny-ImageNet-200 with the greatest $\Delta$ and corresponding top-10 similar samples in $S_{k+1}$ in Appendix 5.

to class "blue" and forget the link to the class "sport car". Moreover, in Appendix 4, we show that increasing $\alpha$ amplifies these trends, demonstrating that stronger subpopulation shifting intensifies the model's reminder of old samples. Intuitively, exposing the model to more new examples strengthens their pull on related old representations. This behavior aligns with the patterns in Figure 1.

However, we also observe two unexpected phenomena: (1) In Figure 5a, even in the highest-similarity quantile, some old data become safer with negative $\Delta$. (2) In BERT, Figure 5c shows that in the target superclass, with increased NTK-similarity, the Privacy Déjà Vu Effect is unchanged or even slightly weakened, which is a reversed pattern compared to the image data on ViT.

To further explore these anomaly patterns, we examine the details of sample similarity levels. Inspired by (Carlini et al., 2022a), who have already hinted that duplicating an image in the training set lowers LiRA's accuracy for each of the duplicated samples. We therefore hypothesize that extremely high similarity between $S_{k+1}$ and a vulnerable point in $S_k$ can act as a privacy shield.

To probe the idea, we run a controlled duplication test. For model $f_k$, we first identify the most and least vulnerable sample (samples with the greatest and smallest risk $R_k$). The most sensitive sample has $R_k = 129.36$ in ViT and $R_k = 91.74$ in BERT. Then we pick the top-100 samples in $\mathcal{D}$ (the entire Tiny-ImageNet-200 or IMDb training set) that are similar to the most sensitive sample in $S_k$, and sort them into 10 shards, $\{\text{Shard}_{i,k+1}|i = 0, ..., 9\}$. $\text{Shard}_{0,k+1}$ is the most similar shard consisting of the 10 most similar samples. Each shard is used for fine-tuning in step k+1 instead, i.e., $S_{k+1} \in \{\text{Shard}_{i,k+1}|i = 0, ..., 9\}$. Similarly, we also pick the top-100 samples in $\mathcal{D}$ most similar to the least vulnerable sample and conduct the fine-tuning.

Figure 6a shows that the most similar shard will "shield" the old most sensitive sample, and then the "shield" becomes weaker when shards consists of less similar samples. It explains why there are some samples with negative $\Delta$ existing in quantile 10 in Figure 5a. In contrast, Figure 6b shows that for the lowest privacy-risk sample, its less similar samples in $S_{k+1}$ will increase the privacy risk more. This explains why the positive $\Delta$ is small in Figure 5c.

Why are the bursty privacy-risk changes in Figure 5a for ViT models not observed in Figure 5c for BERT models? We suspect that the IMDb data has low diversity within the superclass, and the distributional shifts are not so obvious between fine-tuning steps. As a result, the step k+1 uses a similar dataset to step k, leading to small changes in privacy risks. Table 1 partially supports our conjecture. The IMDb batches in target superclasses have much higher NTK-similarity.

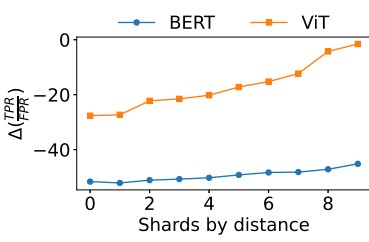

(a) Most sensitive sample in $f_k$

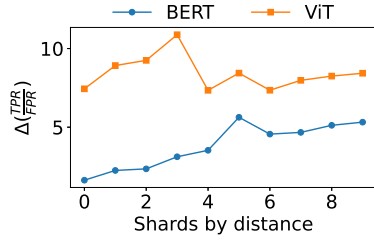

(b) Least sensitive sample in $f_k$

Figure 6: The most similar shard will "shield" the old most sensitive sample. This partially explains two abnormal phenomena. X-axis is the shard ID. The leftmost index means $S_{k+1}$ consists of the most similar shard. The most sensitive sample has $R_k = 129.36$ in ViT and $R_k = 91.74$ in BERT. The least sensitive sample has $R_k = 0.99$ in ViT and $R_k = 0.986$ in BERT. We visualize the samples in the Appendix 6.

| Dataset (model) | Target superclass | Other superclass |
|---|---|---|
| Tiny-ImageNet-200 (ViT) | $0.4484 \pm 0.0175$ | $0.3731 \pm 0.0188$ |
| IMDb (BERT) | $0.8791 \pm 0.0094$ | $0.1224 \pm 0.0172$ |

Table 1: Normalized NTK-similarity (mean $\pm$ std) for target vs. other superclasses. The IMDb batches in target superclasses have much higher NTK-similarity, which partially implies the IMDb data has low diversity within superclasses.

In summary, the Privacy Déjà Vu Effect is rather a local phenomenon: a few similar samples in the new batch trigger the dramatic privacy risk changes of a few old samples. However, as shown in Figure 1, the model is still forgetting legacy data generally. Whether an old example becomes riskier or safer depends chiefly on *feature-level* neighbours it gains in the new round and on the intrinsic complexity of the dataset. Some may question that this may be due to the model re-learning the old sample. However, the old sample will not appear in $S_{k+1}$. Moreover, in our Appendix, we show that both $f_k$ and $f_{k+1}$ do not overfit on the sample with the greatest $\Delta$. Thus, the effect cannot be concluded as re-memorization. Meanwhile, the relationship is not monotonic: If the old sample is sensitive, initially its risk drops as similarity of newcomer increases—new examples "cover" it better—until a similarity threshold beyond which additional resemblance no longer helps; if the old sample is safe, fine-tuning with similar data raises its privacy risk, but this effect also stop increasing past a certain threshold. Pinpointing those similarity thresholds depends on the model's capacity and the complexity of the dataset, and how to identify them remains an open challenge.

# 6 CONCLUSION

Our study of the Privacy Déjà Vu Effect reveals critical implications for privacy protection in continual fine-tuning systems. We summarize our observations and then propose possible solutions.

**Reassessing "Safe" Legacy Data.** Many studies sugguest that catastrophic forgetting can ease the risk of old samples being breached by privacy attacks (Wang et al., 2025). However, we observe that fine-tuning on new data with high feature-level similarity can *rehabilitate* sensitive aspects of old samples, triggering renewed privacy exposure. This Privacy Déjà Vu Effect means that prioritizing the protection of only new data can leave old data unexpectedly vulnerable; privacy mechanisms must therefore guard across all fine-tuning rounds, not just the most recent one.

**Open Questions.** While our work focuses on domain-incremental fine-tuning, it remains an open question whether the Privacy Déjà Vu Effect manifests in other fine-tuning paradigms, where the label space also evolves. Meanwhile, constrained by the heavy computational cost, we can only show the existence of the effect in two rounds of fine-tuning on representative models. However, our initial observations indicate that other models and additional rounds of fine-tuning are likely to have this effect as well, which will be verified in future work.

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

# A APPENDIX

## A.1 ILLUSTRATION OF FINE-TUNING MODELS AND ESTIMATION OF PRIVACY SCORE

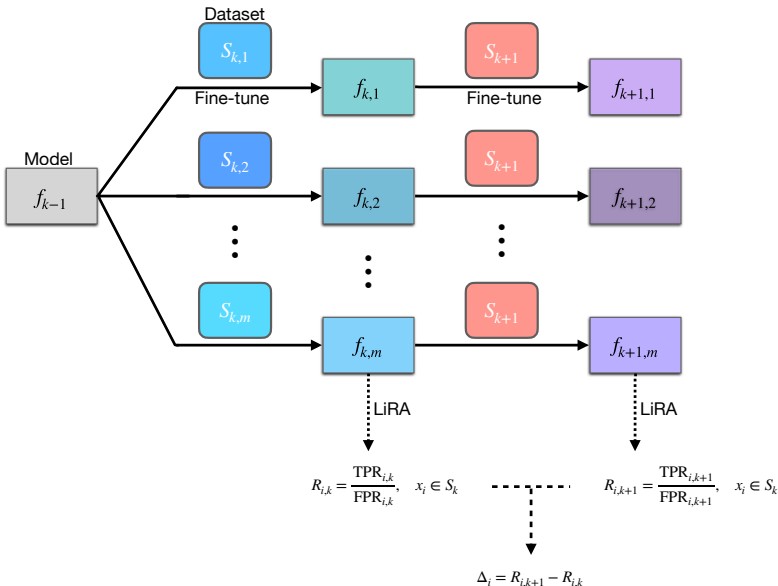

Figure 7: Fine-tune models and estimate per-sample privacy scores difference $\Delta_i$. $S_{k,m} \subset S_k$, $S_k \cap S_{k+1} = \phi$.

## A.2 NTK-SIMLARITY

Let $g_k(x) = \nabla_\theta f_k(x)$ be the gradient of the fine-tuned round-$k$ model with respect to input $x$ and define

$$K(x, x') = \langle g_k(x), g_k(x') \rangle.$$

For every $x \in S_k$ we compute its mean similarity to the new set:

$$\bar{K}(x) = \frac{1}{|S_{k+1}|} \sum_{x' \in S_{k+1}} K(x, x').$$

## A.3 SSIM AND $\Delta$: VARIOUS $\alpha$

We show the correlation between SSIM-similarity and $\Delta$ in various $\alpha$ settings. Figure-8 to 10 show no significant correlations.

## A.4 NTK-SIMILARITY AND $\Delta$: VARIOUS $\alpha$

We show the correlation between NTK-similarity and $\Delta$ in various $\alpha$ settings. Figure-11 to 13 show apparent correlations.

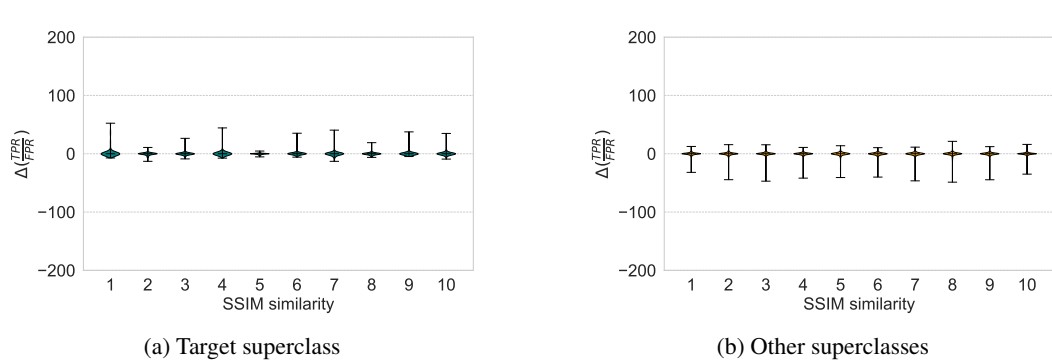

(a) Target superclass           (b) Other superclasses

Figure 8: SSIM vs. $\Delta$: $\alpha = 0.2$

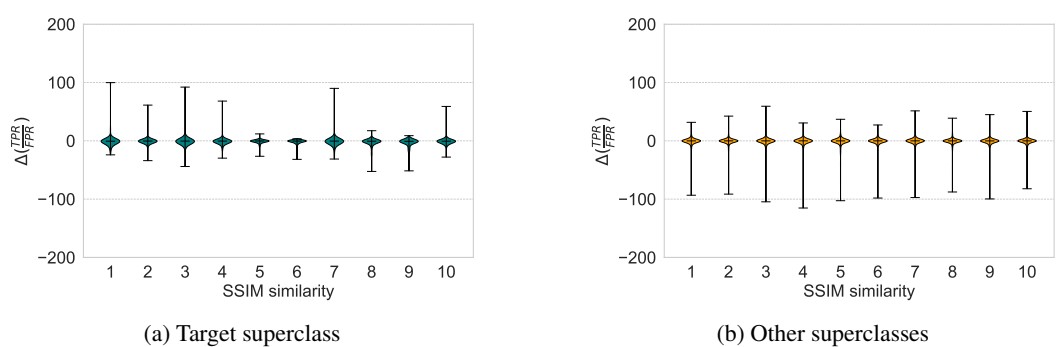

(a) Target superclass           (b) Other superclasses

Figure 9: SSIM vs. $\Delta$: $\alpha = 0.4$

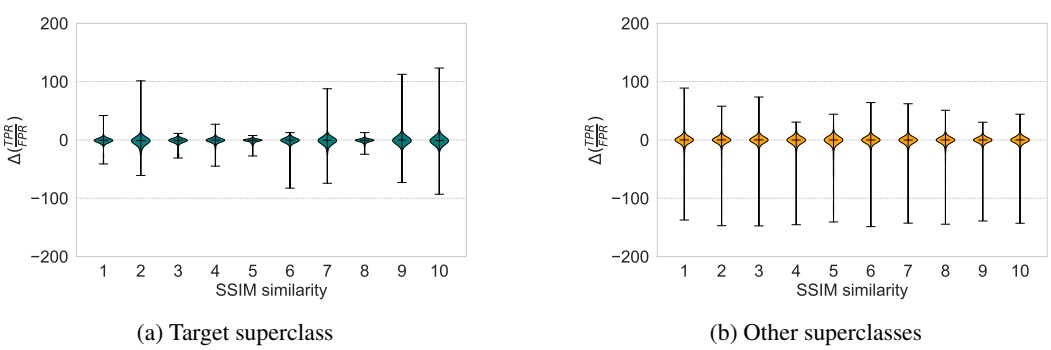

(a) Target superclass           (b) Other superclasses

Figure 10: SSIM vs. $\Delta$: $\alpha = 0.6$

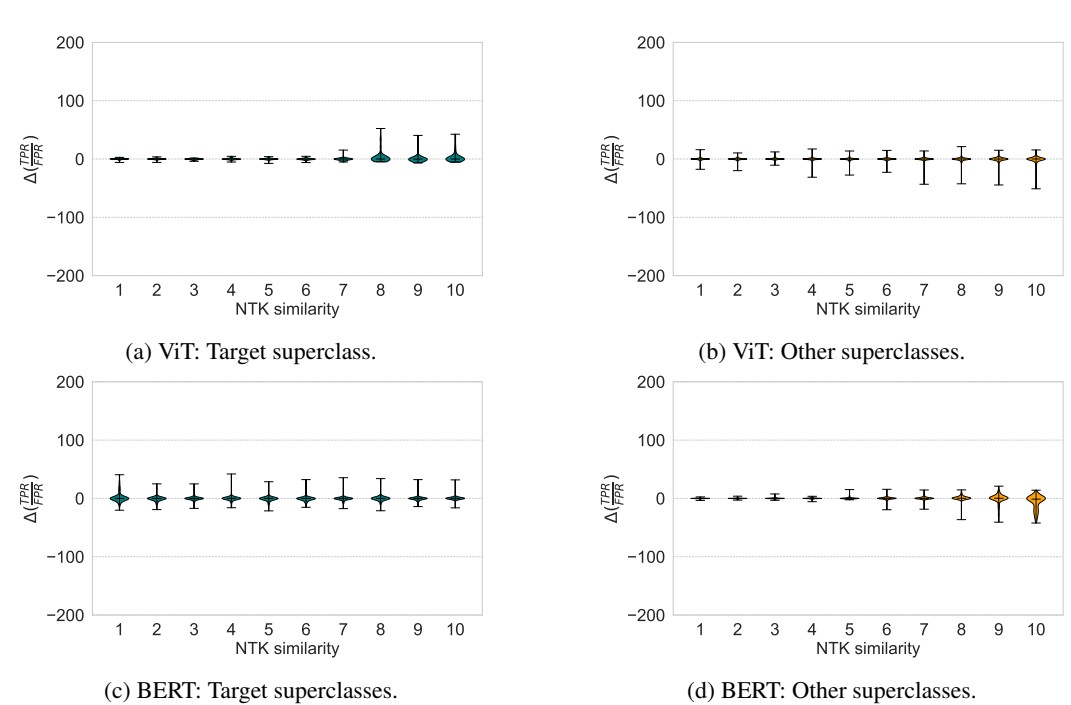

Figure 11: Violin graph of each quantile. ViT: $\alpha = 0.2$; BERT: $\alpha = 0.4$.

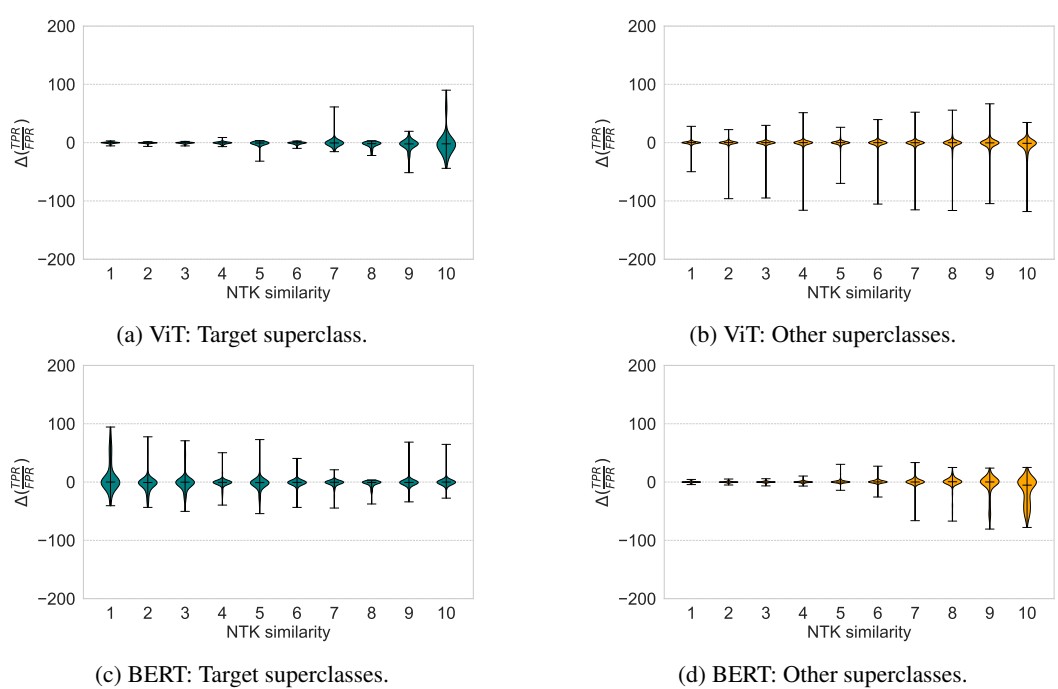

Figure 12: Violin graph of each quantile. ViT: $\alpha = 0.4$; BERT: $\alpha = 0.7$.

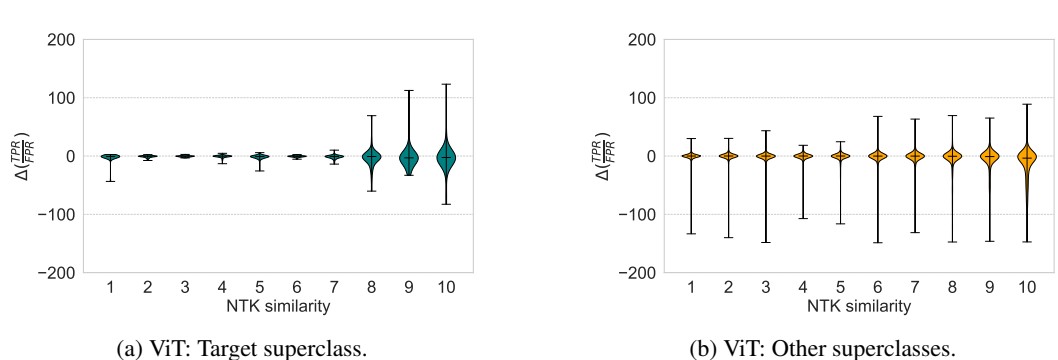

(a) ViT: Target superclass.

(b) ViT: Other superclasses.

Figure 13: Violin graph of each quantile. ViT: $\alpha = 0.6$.

## A.5 Visualization of Samples with the Greatest $\Delta$

We show the old samples with the greatest $\Delta$ (circled by red rectangle), and their nearest neighbors with the greatest NTK-similarity in this section. Figure 14 shows the old sample from the target superclass "mammal". We have tested the sample whose privacy risk increases most, e.g., the circled sample in Figure 14, on all models and found that $f_k$'s accuracy is 89.3% and $f_{k+1}$'s is 87.5%. In comparison, the average fine-tuning accuracy of fine-tuning set on the models is 97.3%. Thus, the most sensitive sample appears not to be overfitted by $f_k$ and $f_{k+1}$. Furthermore, models seem to perform worse after seeing more new samples.

Figure 14 shows the old sample from one of the other superclasses, "instrumentality".

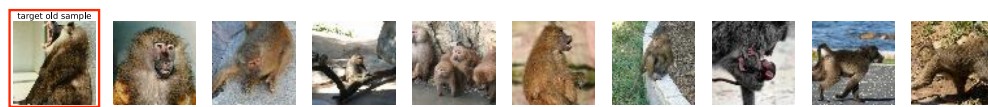

Figure 14: Sample from target superclass with greatest $\Delta = 147.3$ and the most similar samples in $S_{k+1}$. Target superclass = "mammal".

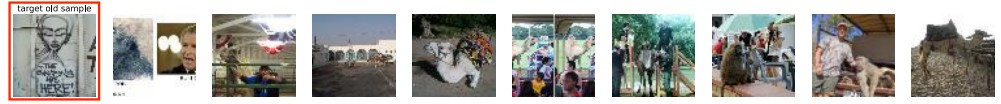

Figure 15: Sample from other superclass with greatest $\Delta = 92.2$ and the most similar samples in $S_{k+1}$. Target superclass = "mammal". Superclass of sample = "instrumentality".

## A.6 Visualization of the Top-100 Most NTK-similar Samples of the Target Old Sample

We show the old samples with the greatest and smallest $R_k$ (circled by red rectangle) in this section. Figure 16 shows the sample with the greatest $R_k$ and its top-100 most NTK-similar samples. Figure 17 shows the sample with the smallest $R_k$ and its top-100 most NTK-similar samples.

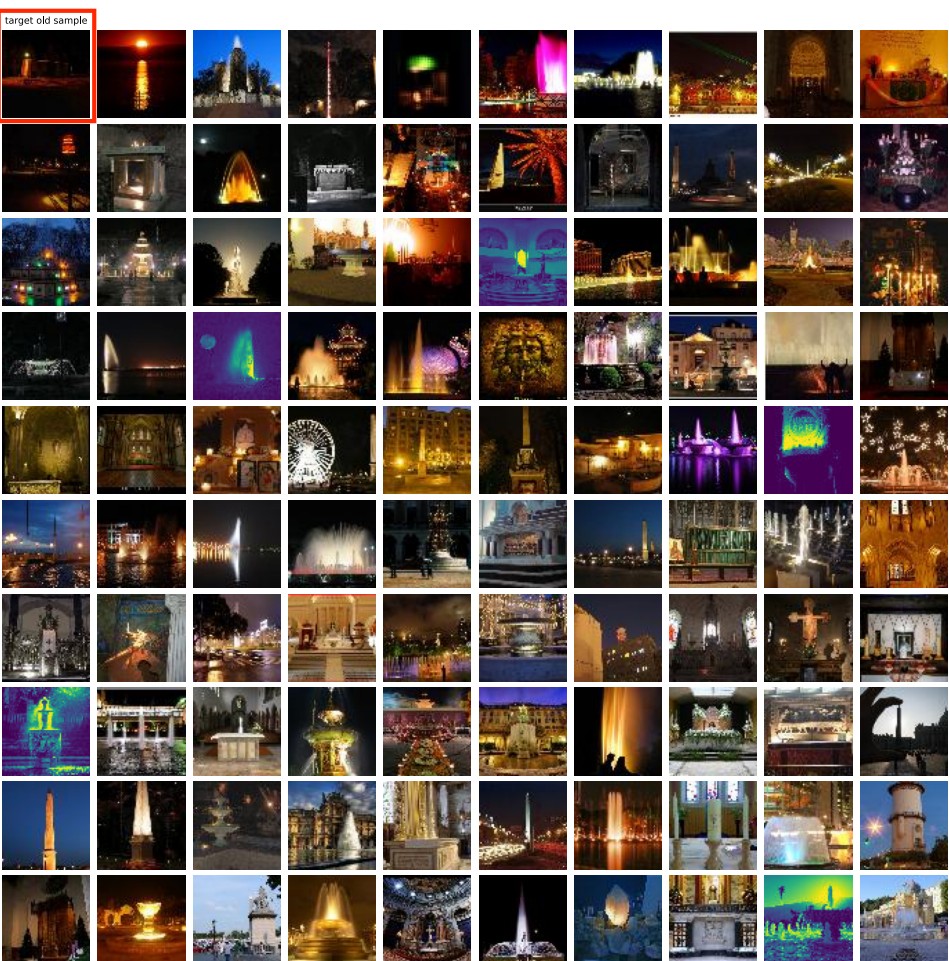

Figure 16: Visualized most sensitive old sample (top left). Figures from left to right have smaller similarity. Some may not be visually similar, but with high NTK-similarity.

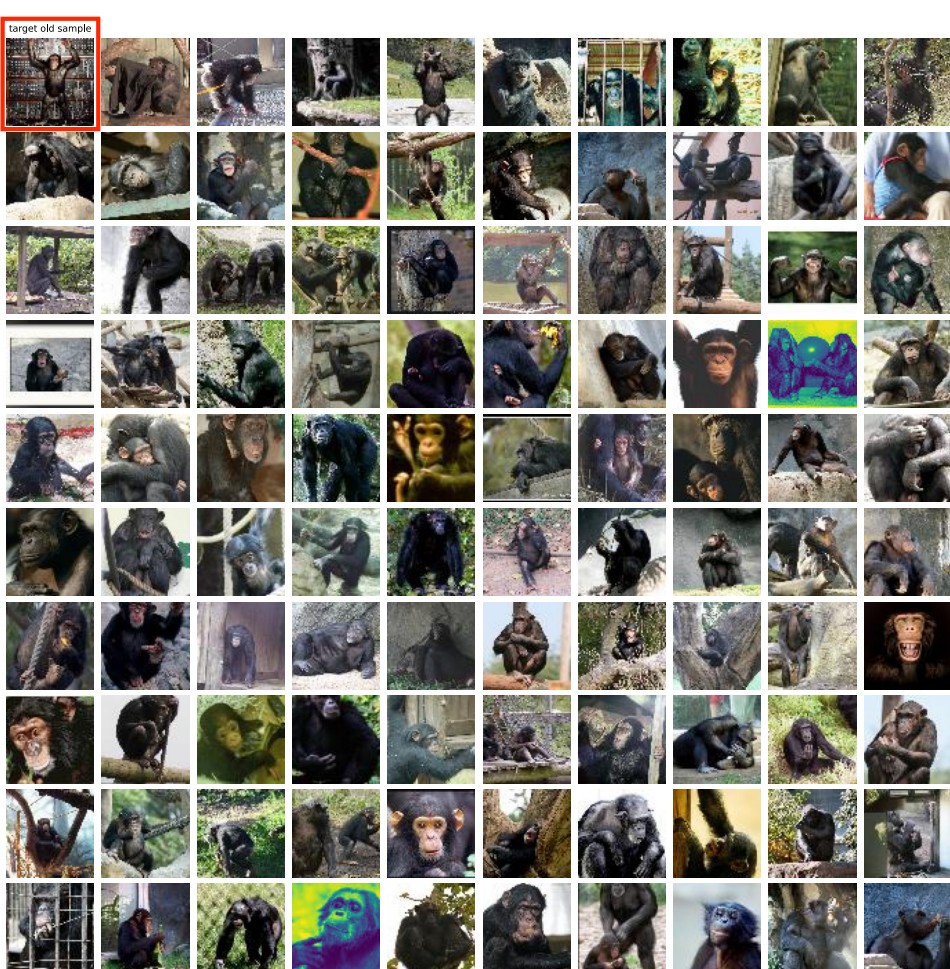

Figure 17: Visualized most vulnerable old sample (top left). Figures from left to right have smaller similarity. The bottom right figure has the least NTK-similarity.

