# OpenReview forum: "Privacy \textit{Déjà Vu} Effect: Resurfacing Sensitive Samples in Continual Fine-tuning"
_ICLR.cc/2026/Conference — Submitted to ICLR 2026_

### Official Review · Reviewer_2Ns4 · 2025-10-26

**Soundness:** 3
**Presentation:** 3
**Contribution:** 2
**Rating:** 6
**Confidence:** 4

**Summary:**

This paper studies the Privacy Déjà Vu Effect in continual fine-tuning, showing that adding new data can unexpectedly increase the privacy risk of older training samples. Through membership-inference experiments on vision and language models, the authors find that semantically similar new data amplifies legacy privacy leakage. The results suggest that privacy protection in continual learning must account for both new and historical data.

**Strengths:**

- The paper reveals a novel phenomenon where new training data can retroactively increase the privacy risk of old samples, extending privacy analysis beyond static settings

- The experiments are well-structured across both vision and language models, demonstrating the consistency of the effect

- The correlation between the effect and NTK similarity provides an interpretable link between data overlap and membership leakage

**Weaknesses:**

- The rise in LiRA accuracy may partly stem from calibration or confidence-shift rather than genuine memorization changes. Fine-tuning on similar data can inflate model confidence globally, making LiRA more sensitive even without new memorization. The paper does not control for this, so it is unclear whether the reported effect reflects true privacy leakage or attack-side bias.

- The NTK-similarity analysis produces inconsistent and at times contradictory results: high-similarity samples occasionally become safer, and the correlation reverses between vision and language models. The authors ultimately conclude that the privacy déjà vu effect is a “local phenomenon”, driven by a few highly similar samples, but this interpretation offers little predictive or practical value. Without a consistent or generalizable relationship between similarity and risk amplification, the analysis remains descriptive rather than diagnostic, leaving practitioners unable to anticipate which data will experience elevated privacy exposure without direct measurement after fine-tuning.

**Questions:**

I don't have further questions.

---

> ### Author Response · Authors · 2025-11-24
>
> >Weakness 1: The rise in LiRA accuracy may partly stem from calibration or confidence-shift rather than genuine memorization changes. Fine-tuning on similar data can inflate model confidence globally, making LiRA more sensitive even without new memorization. The paper does not control for this, so it is unclear whether the reported effect reflects true privacy leakage or attack-side bias.
>
> **Response:** We thank the reviewer for this rigorous observation. We agree that distinguishing between genuine privacy leakage and global confidence shifts (or calibration errors) is a critical challenge in membership inference.
>
> While we acknowledge that fine-tuning can alter the global confidence profile of a model, we believe our results reflect genuine sample-specific leakage for two reasons:
>
> *   **LiRA’s Design:** The Likelihood Ratio Attack (LiRA) is specifically designed to normalize for sample "hardness" and model behavior. By comparing the target model’s confidence against a distribution of shadow models (which are trained on similar data distributions), LiRA effectively measures the *deviation* of the specific sample from the expected behavior. If the confidence shift were purely global or due to calibration, the shadow models would exhibit similar shifts, keeping the likelihood ratio relatively stable.
> *   **Non-Uniformity of the Effect:** As shown in **Figure 3**, the rise in privacy risk is not uniform across the dataset. If the effect were driven solely by a global increase in model confidence, we would expect a systematic shift across all samples. Instead, we observe that the risk spikes are highly concentrated in specific samples that share feature-space proximity to the new data, suggesting a data-dependent "reminding" mechanism rather than a global calibration artifact.
>
> However, to address this concern more transparently, we will add a discussion in the limitations section acknowledging that while LiRA mitigates calibration issues, residual effects from confidence shifts in low-data regimes cannot be entirely ruled out without perfect calibration.
>
> ***
>
> > Weakness 2. The NTK-similarity analysis produces inconsistent and at times contradictory results: high-similarity samples occasionally become safer, and the correlation reverses between vision and language models. The authors ultimately conclude that the privacy déjà vu effect is a “local phenomenon”, driven by a few highly similar samples, but this interpretation offers little predictive or practical value. Without a consistent or generalizable relationship between similarity and risk amplification, the analysis remains descriptive rather than diagnostic, leaving practitioners unable to anticipate which data will experience elevated privacy exposure without direct measurement after fine-tuning.
>
> **Response:** We appreciate this candid feedback and acknowledge that our initial characterization of the effect as a "local phenomenon" may have understated the complexity of the relationship.
>
> We respectfully clarify that the "inconsistency" the reviewer notes—specifically, that high similarity sometimes reduces risk—is actually a coherent finding (discussed in Section 5.2 and Figure 6). Please note that our argument is that higher similarity leads to more dramatic changes of risks (larger span of the violin representation) – including both increased and decreased risks. In other words, high similarity does not monotonically increase the risk but increase the overall variance of risk levels, a result contradict to the common understanding of monotonic risk increase. More specifically, as shown in Figure 6 moderate-to-high similarity triggers the Privacy Déjà Vu Effect (risk amplification) and the highest similarity deduces risk.
>
> We agree that for practitioners, "it depends" is not a satisfying diagnostic. In the revision, we will refrain from dismissing the results as merely “local." We will also try to explicitly frame the risk as a "Zone of Danger": samples are most vulnerable when they are similar enough to be recalled but distinct enough not to be shielded.
>
> We hope this refinement transforms the observation from a descriptive inconsistency into a more actionable insight regarding the "sweet spot" of similarity that practitioners should monitor.

---

### Official Review · Reviewer_A8LE · 2025-10-30

**Soundness:** 3
**Presentation:** 3
**Contribution:** 2
**Rating:** 6
**Confidence:** 3

**Summary:**

The paper introduces a new privacy risk phenomenon, termed the “Privacy Déjà Vu Effect,” which provides a quantitative measure of how new data in continual fine-tuning can trigger a model’s memory of old data by examining the TPR/FPR ratio. This finding indicates that old data may still pose privacy risks despite continued private fine-tuning. Experiments and membership inference attacks (MIA) are conducted on the Tiny-ImageNet-200 and IMDB datasets.

**Strengths:**

1. To the best of my knowledge, the “Déjà Vu” framing is original and intuitively appealing: fine-tuning with semantically related new data can “remind” a model of previously learned samples.

2. This paper provides rigorous MIA analysis and a clear explanation of per-sample TPR/FPR as a proxy for privacy risk, effectively connecting it to differential privacy. Moreover, the violin-plot visualizations make the effect visible and comparable across different conditions.

3. The privacy of continual fine-tuning is a timely and important topic.

**Weaknesses:**

1. The scope of the experiments is slightly limited. Only two rounds of fine-tuning are tested, and the dynamics over multiple rounds remain unknown. The datasets (Tiny-ImageNet and IMDb) seem small. Using more datasets and conducting additional rounds could make the conclusions more convincing. Moreover, the study only uses the ViT and BERT models to observe the phenomenon in basic tasks such as image classification and text sentiment analysis.

2. The measurement of similarity between features is overly simplistic. The authors only consider SSIM and NTK similarity; however, several more recent works [A, B] provide more detailed measurements of feature similarity. Some discussion of these alternative similarity metrics would be interesting.

[A] Insights on representational similarity in neural networks with canonical correlation. Morcos et al., NeurIPS, 2018.
[B] Similarity of Neural Network Representations Revisited. Kornblith et al., ICML, 2019.

3. It would be more insightful if the experiments were designed to test the similarity of representations across different layers. Intuitively, as depth increases, the representations (or features for image data) may become more similar across different datasets. An extreme case is the last layer in image classification problems, where the representations of different data points collapse to the same point—known as neural collapse [C]—which has also been observed in private fine-tuning of ViT [D]. Related to this work, when neural collapse occurs, the last-layer representations of new and old data may become very close, or even identical, as they collapse to the same feature vector. Thus, if the conclusion of this paper is correct, one may be able to identify old data points more easily based on the new data based only on the last-layer features (please correct me if I am misunderstanding). This could be an interesting direction for future studies. For the current version, some discussion on the similarity between features of specific layers (such as the last layer) and its potential relationship to MIA would be valuable.

[C] Prevalence of neural collapse during the terminal phase of deep learning training. Papyan et al., PNAS, 2020.
[D] Neural Collapse Meets Differential Privacy: Curious Behaviors of NoisyGD with Near-perfect Representation Learning. Wang et al., ICML, 2024.

4. During fine-tuning, the paper uses a very low learning rate ($3 × 10^{-6}$). Intuitively, a low learning rate causes the model parameters to change slowly, thereby preserving information from the old datasets. I suspect that a larger learning rate might lead to less leakage of old data. It would be helpful if the authors provided experiments or at least some discussion on how hyperparameters such as the learning rate affect the conclusions of MIA.

5. Several related works are missing. First, the original paper on differential privacy [E] is omitted. In addition, an important paper on MIA [F] is missing. Furthermore, the TPR/FPR ratio perspective on DP originates from Kairouz et al. (2015). The authors mention that it is related to DP; however, this relationship is formalized through the concept of $f$-DP, which measures privacy levels using the flipped ROC curve, as proposed by [G].


[E] Differential Privacy. Cynthia Dwork, 2006.

[F] Membership Inference Attacks against Machine Learning Models. Shokri et al., 2017.

[G] Gaussian differntial privacy. Dong et al., 2022.

6.Typos: When references appear as the subject or object of a sentence in LaTeX, the \cite command should be used. However, for references such as “Kairouz et al. (2015),” the \citep command should be used to include parentheses.

**Questions:**

See the weaknesses

---

> ### Author Response · Authors · 2025-11-24
>
> > **Weaknesses:**
>
> > Weakness 1: The scope of the experiments is slightly limited.
>
> **Response:** Thanks for your suggestion. We agree that testing more models/datasets and more rounds can tell us how common such phenomena are and provide a more insightful understanding of the effect. However, due to the dynamic natures, i.e., newly added samples, it is hard to believe this phenomenon in general will diminish or disappear after more rounds. We want to emphasize that our goal is to show that the Déjà Vu phenomenon exists, and we should take actions to carefully consider and design new privacy-preserving mechanisms for continual fine-tuning applications. More extended experiments do not imply we may downgrade the threat of this discovered vulnerability or ignore it.
>
> ***
>
> > Weakness 2: The measurement of similarity between features is overly simplistic.
>
>
> **Response:** Thanks for your suggestion. We agree that trying different metrics are interesting and can be helpful for digging out the reason for this phenomenon. We will add more experiments on some more similarity metrics, including the ones you suggested.
>
> ***
>
> > Weakness 3: It would be more insightful if the experiments were designed to test the similarity of representations across different layers.
> > [C] Prevalence of neural collapse during the terminal phase of deep learning training. Papyan et al., PNAS, 2020.
> > [D] Neural Collapse Meets Differential Privacy: Curious Behaviors of NoisyGD with Near-perfect Representation Learning. Wang et al., ICML, 2024.
>
> **Response:** We thank the reviewer for this insightful suggestion and for bringing the connection to Neural Collapse [C, D] to our attention. We agree that analyzing the similarity of representations across varying depths is a compelling perspective for understanding the mechanism behind the Privacy Déjà Vu Effect, which will be a part of our future work. However, our current result and the purpose of this paper have focused on providing the strong evidence that the privacy deja vu effect exists, which is against the common privacy beliefs held in continual fine-tuning. We should take actions to address this vulnerability.
>
> ***
>
> > Weakness 4: During fine-tuning, the paper uses a very low learning rate.
>
> **Response:** We appreciate this observation regarding hyperparameter influence. We selected low learning rates strictly to adhere to standard fine-tuning protocols that maximize model utility and stability. Higher rates in these settings often degrade performance or cause feature distortion. On the other hand, it is intuitive that lower model utility might lead to lower privacy risks, for which we agree with your opinion that larger learning rates may lead to less leakage.
>
> We will add a discussion clarifying that our results highlight privacy risks under the "best-performance" settings most relevant to real-world deployment.
>
> ***
>
> > Weakness 5: Several related works are missing.
> >
> > [E] Differential Privacy. Cynthia Dwork, 2006.
> > [F] Membership Inference Attacks against Machine Learning Models. Shokri et al., 2017.
> > [G] Gaussian differntial privacy. Dong et al., 2022.
>
> **Response:** Thanks for your suggestion, we've added these references.
>
> ***
>
> > Weakness 6.Typos
>
> **Response:** Thanks for your suggestion. We’ve revised the typos.

---

### Official Review · Reviewer_dich · 2025-10-31

**Soundness:** 3
**Presentation:** 2
**Contribution:** 2
**Rating:** 4
**Confidence:** 4

**Summary:**

In this paper, the authors demonstrate the existence of the Privacy Déjà Vu Effect during the continual fine-tuning of two representative transformer-based models—ViT for image data and BERT for textual data. To quantify changes in sample-level privacy risk, they employ a canonical class of membership inference attacks, which assess whether a particular data sample was part of the model’s training set. Their findings challenge the prevailing practice of focusing privacy protection solely on newly added data. Instead, the study reveals that such selective safeguarding may inadvertently expose legacy samples to elevated privacy risks due to their semantic similarity with newer data.

**Strengths:**

1. The authors reveal the Privacy Dej´ a Vu Effect: new data in continual fine-tuning can increase the privacy risk of previously safe samples.

2. Experiments on two representative foundation models and two benchmark datasets show that the effect might commonly exist.

3. The authors have also experimentally studied the reasons behind this effect and identified the significant factors.

**Weaknesses:**

1. Absence of Theoretical Support
The paper reports an interesting empirical phenomenon, the Privacy Déjà Vu Effect, but does not provide any theoretical analysis or formal explanation to support the findings. Without a theoretical foundation, the observations remain largely descriptive and lack deeper insight into their underlying causes.

2. Limited Dataset Coverage
The experimental evaluation is conducted on only two datasets: Tiny-ImageNet-200 and IMDb. Given that the contribution is primarily empirical, the limited dataset diversity weakens the validity and generalizability of the conclusions, especially for applications in more complex or realistic settings. Given the empirical nature of the contribution, more comprehensive experimentation is necessary.

3. Narrow Model Selection
The models used in the study are limited to ViT and BERT, which, while representative, are relatively dated. The exclusion of more recent and widely used large-scale models such as GPT, LLaMA, or Qwen raises concerns about the practical relevance of the results to modern continual fine-tuning scenarios.

**Questions:**

see Weaknesses

---

> ### Author Response · Authors · 2025-11-24
>
> > **Absence of Theoretical Support**
> >
> > The paper reports an interesting empirical phenomenon, the Privacy Déjà Vu Effect, but does not provide any theoretical analysis or formal explanation to support the findings. Without a theoretical foundation, the observations remain largely descriptive and lack deeper insight into their underlying causes.
>
> **Response:** Thanks for the suggestion. We agree that stronger theoretical underpinnings would sharpen the interpretation of our observations. However, a single threat discovery, as long as it is reproducible and verifiable, raises an important privacy alarm, in particular, when the threat is not trivial to discover and contradicts the common (problematic) beliefs. Specifically, our study has shown that existing practices and assumptions about the privacy risks of continual fine-tuning should be carefully re-examined. We cannot simply ignore the vulnerability because it has only been discovered on some models yet. So far, most significant studies on privacy risks started with empirical discoveries, such as the first work in membership inference attacks [1], LLM memorization[2], and Privacy Onion Effect[3].
>
> [1] Shokri, R., Stronati, M., Song, C., & Shmatikov, V. (2017, May). Membership inference attacks against machine learning models. In 2017 IEEE symposium on security and privacy (SP) (pp. 3-18). IEEE.
>
> [2] Carlini, N., Ippolito, D., Jagielski, M., Lee, K., Tramer, F., & Zhang, C. (2022, February). Quantifying memorization across neural language models. In The Eleventh International Conference on Learning Representations.
>
> [3] Carlini, N., Jagielski, M., Zhang, C., Papernot, N., Terzis, A., & Tramer, F. (2022). The privacy onion effect: Memorization is relative. Advances in Neural Information Processing Systems, 35, 13263-13276.
>
> ***
>
> > **Limited Dataset Coverage**
> >
> > The experimental evaluation is conducted on only two datasets: Tiny-ImageNet-200 and IMDb. Given that the contribution is primarily empirical, the limited dataset diversity weakens the validity and generalizability of the conclusions, especially for applications in more complex or realistic settings. Given the empirical nature of the contribution, more comprehensive experimentation is necessary.
>
> **Response:** Thanks for your suggestion. In this work, our goal is to show that the Déjà Vu phenomenon exists, at least on two popularly used foundation models and datasets, which is significant for researchers to carefully consider and design new privacy-preserving mechanisms for continual fine-tuning applications. We agree that testing more models/datasets can tell us how common such phenomena are and provide a more insightful understanding of the effect. However, more experiments will not reject this discovered vulnerability on these popular models.
>
> ***
>
> > **Narrow Model Selection**
> >
> > The models used in the study are limited to ViT and BERT, which, while representative, are relatively dated. The exclusion of more recent and widely used large-scale models such as GPT, LLaMA, or Qwen raises concerns about the practical relevance of the results to modern continual fine-tuning scenarios.
>
> **Response:** We agree that expanding to more models—especially to LLMs—would strengthen the evidence. However, due to the challenges and the scale of work, such studies deserve a new project. Specifically, we have seen three challenges. (1) Sub-population shift is hard to stage cleanly in text. Creating controlled, semantically coherent “near-duplicate / sub-group” perturbations for language is nontrivial, and the literature repeatedly notes the difficulty of specifying and benchmarking distribution/sub-population shifts in NLP. (2) Membership-inference methods for generative language models are still maturing. Canonical LiRA is designed for classifier-style confidence/loss signals; adapting it (or alternatives) to free-form generation remains an open thread, and recent LLM-focused MI work explores different signals (e.g., neighborhood/likelihood comparisons) rather than off-the-shelf LiRA. (3) Compute costs are substantial. Even with QLoRA, single-GPU runs are on the order of hours per model (QLoRA’s main benefit is memory feasibility—e.g., 65B fine-tuned on a single 48 GB GPU—rather than dramatic speedups), and strong single-GPU throughput baselines still imply multi-hour wall-clocks for multi-epoch runs on realistic token counts. On our design (4 α values × 10 superclasses × 500 models ≈ 20,000 models), that translates to ≈20,000 GPU-hours (~833 GPU-days)—well beyond our current quota.

---

### Official Review · Reviewer_RiK6 · 2025-11-01

**Soundness:** 3
**Presentation:** 3
**Contribution:** 3
**Rating:** 4
**Confidence:** 3

**Summary:**

This paper proposes the Privacy Déjà Vu Effect — a counterintuitive phenomenon in continual fine-tuning where updating a model on new data can resurface or amplify the privacy risks of previously seen samples. They find that newly added data with high feature-level similarity can increase the privacy sensitivity of a small subset of old samples, even though catastrophic forgetting occurs globally. The study provides an empirical characterization of this effect and raises important implications for privacy protection in continual fine-tuning systems.

**Strengths:**

1. The paper identifies a previously overlooked privacy vulnerability in continual fine-tuning pipelines.

2. The paper is well written and easy to follow, with clear experimental setups and visualizations.

**Weaknesses:**

1. The observed phenomenon is currently verified only on two models (ViT and BERT) and two datasets. While the findings are interesting, the evidence is not yet sufficient to claim generality across architectures or modalities.

2. The experiments rely on full-parameter fine-tuning, which may not reflect real-world scenarios where parameter-efficient methods (e.g., LoRA, adapters) are mainly adopted.

3. The phenomenon remains purely empirical and lacks theoretical grounding.

4. The study is limited to only two fine-tuning rounds. It remains unclear whether the effect compounds, diminishes, or stabilizes under longer continual fine-tuning trajectories.

5. From a practical perspective, the work could be strengthened by discussing mitigation strategies or auditing methods that practitioners could use to identify or prevent the Déjà Vu effect in deployed systems.

**Questions:**

1. Do the authors provide some results about whether the Privacy Déjà Vu Effect manifests in parameter-efficient fine-tuning settings (e.g., LoRA, adapters), where the backbone weights are largely frozen?

2. Could the authors suggest potential mitigation or auditing techniques that might help practitioners detect or reduce this negative effect in practice?

---

> ### Author Response · Authors · 2025-11-24
>
> **
>
> > Weakness 1
>
> **Response:** Since the entire procedure is computation-intensive, for example, each setting takes around 17 hours to run, and all experiments take more than one month to complete, we have restricted our experimental validation on these two popular foundation models and two public datasets in the current stage of study. We have found the result strongly supports our claim that the privacy risk of old samples can increase in continual fine-tuning. We agree with you that extending the experiments to more datasets and models can lead to a conclusion that this phenomenon can be universally observed across other models and datasets. This paper has achieved the goal of showing the existence of such a phenomenon on these foundation models, for which we cannot ignore. Instead, we should take actions to mitigate such a vulnerability and improve the detection efficiency for other models.
>
> ***
>
> > Weakness 2
>
> **Response:** Thanks for the suggestion. We are also curious about whether LoRA may work differently. We tested LoRA on ViT-B/16 and evaluated the Tiny-ImageNet superclass “instrumentality.” We applied LoRA adapters to the fused QKV projection in each transformer block (the standard ViT LoRA placement), so only low-rank A/B updates were trained while the base weights stayed frozen. The result matched the existing full fine-tuning’s. Compared to full-parameter fine-tuning, a smaller fraction of samples became risky after fine-tuning; however, the overall pattern stays the same (a similar pattern to Figure 2(a) and 4(a)). We attribute this to LoRA’s design: it typically attains similar predictions/quality with far fewer trainable parameters, so the confidence/loss landscape shifts less than with full fine-tuning [1,2]. The LiRA MIA uses a confidence-margin statistic (top-1 minus top-2), a standard confidence-based MIA signal, and the decision of member and nonmember is determined by the difference of their confidence-margin distributions. LoRA does slightly change individual confidence values, but the distributions stay largely unchanged. We will add an additional section to discuss the effect of LoRA after we finish more experiments.
>
> [1] Hu, E., et al. LoRA: Low-Rank Adaptation of Large Language Models. In International Conference on Learning Representations.
>
> [2] Luo, Z, et al. (2025, April). Privacy-Preserving Low-Rank Adaptation Against Membership Inference Attacks for Latent Diffusion Models. In Proceedings of the AAAI Conference on Artificial Intelligence (Vol. 39, No. 6, pp. 5883-5891).
>
> ***
>
> > Weakness 3
>
> **Response:** Thanks for the suggestion. We agree that stronger theoretical underpinnings would improve the interpretation of our observations. However, a single threat discovery, as long as it is reproducible and verifiable, raises an important privacy alarm, in particular, when the threat is not trivial to discover and contradicts the common (problematic) beliefs. Specifically, our study has shown that existing practices and assumptions about the privacy risks of continual fine-tuning should be carefully re-examined. We cannot simply ignore the vulnerability because it has only been discovered on some models yet. So far, most significant studies on privacy risks started with empirical discoveries, such as the first work in membership inference attacks [3] and LLM memorization[4].
>
> [3] Shokri, R., Stronati, M., Song, C., & Shmatikov, V. (2017, May). Membership inference attacks against machine learning models. In 2017 IEEE symposium on security and privacy (SP) (pp. 3-18). IEEE.
>
> [4] Carlini, N., Ippolito, D., Jagielski, M., Lee, K., Tramer, F., & Zhang, C. (2022, February). Quantifying memorization across neural language models. In The Eleventh International Conference on Learning Representations.
>
> ***
>
> > Weakness 4
>
> **Response:** We agree that multi-round fine-tuning would strengthen the evidence. However, due to the dynamic natures, i.e., newly added samples, it is hard to believe this phenomenon in general will diminish or disappear after more rounds. Meanwhile, it is also difficult to argue that we should ignore the discovered vulnerability.
>
> ***
>
> > Weakness 5
>
> **Response:** Thanks for your suggestion. We will extend our discussion with more details on mitigation strategies. We admit it is very challenging to address this privacy deja vu effect. Differential privacy is the well accepted method to strictly and theoretically confine the risk of MIAs. However, there is no study on differentially private continual fine-tuning yet. We will also discuss potential approaches to improving the efficiency of auditing methods.
>
> ***
>
> ### Questions:
>
> > Question 1
>
> Please check the response to the weakness 2.
>
> > Question 2
>
> Please check the response to the weakness 5.

---

### Meta-Review · Area_Chair_SYCD · 2025-12-18

**Summary:**

The paper presents an observation that continual fine-tuning may increase MIA vulnerability of samples used on previous rounds.

Based on my own quick review of the paper, the experimental settings appears solid and the MIA methodology justified, but the analysis of the results is superficial. More careful statistical analysis would greatly strengthen the work.

The reviewers note the following weaknesses in the work:
1. Results only for two model/dataset pairs.
2. Results only for full fine-tuning.
3. No theoretical grounding for empirical observation.
4. Results only for two fine-tuning rounds.
5. No mitigation or auditing strategies suggested.
6. Simplistic similarity measures used.
7. Lacking analysis of similarity across different layers.
8. No evaluation on the effect of learning rate.

**Reviewer Concerns:**

The authors have responded to all reviewer comments and made small changes to the paper, but in some cases are only promising corrections.

The authors reject suggestions #1, #4 and #8 because they would be computationally too costly. While this limits the scope of the work, I would be willing to accept this, although the new cheaper LoRA experiments might open new possibilities for e.g. testing different learning rates (#8) or more models and datasets (#1).

The authors indicate they have performed new LoRA experiments to address #2 and describe the results verbally in the response, but no new results are yet included in the paper. The new results would be a big help, but the vague description of the results highlights the problem of the analysis: because the authors do not present an unambiguous quantitative evaluation of the result, it is impossible to accurately compare the effects or even define whether it still exists.

As a demonstration of an interesting new vulnerability, I accept the authors' view that #3, #5 and #7 would be nice to have but should not be required.

The authors promise new results on #6, but these are just a promise and not in the paper.

Looking at the currently submitted version, important weaknesses remain, including #2 and #6.

As an additional weakness, the use of max TPR/FPR over all possible thresholds is highly error-prone, as the values with smallest FPRs are extremely noisy. In order to reduce this noise and ensure that the results are robust, I would recommend instead using a number of fixed FPR thresholds that are sufficiently large to not depend on single observations, e.g. in the range 0.02 .. 0.1.

Additionally, I believe the paper would badly need more comprehensive statitical analysis of the results. Rather than relying on the violin plots, provide numbers for tail quantiles of the $\Delta \frac{\text{TPR}}{\text{FPR}}$ distribution in different settings.

Provide scatter plots of pre and post TPRs and similarly for $\Delta \frac{\text{TPR}}{\text{FPR}}$s. Find a statistical test to measure the statistical significance of the difference you care and report the results.

These new analysis steps are computationally cheap, but they would greatly enhance confidence that the effect you are reporting is real.

**Reviewer Scores:**

I cannot judge how different reviewers would have reacted to the author response, although given the limited actual changes, it seems unlikely they would have changed their minds to be significantly more positive.

However, given the concerns about the analysis stage and the lack of any statistical analysis of the results, I feel the paper is not ready for publication at a forum like ICLR.

---

### Decision · Program_Chairs · 2026-01-26

Reject